# Copper Ferrite Nanoparticles Synthesized Using Anion-Exchange Resin: Influence of Synthesis Parameters on the Cubic Phase Stability

**DOI:** 10.3390/ma16062318

**Published:** 2023-03-14

**Authors:** Svetlana Saikova, Aleksandr Pavlikov, Denis Karpov, Aleksandr Samoilo, Sergey Kirik, Mikhail Volochaev, Tatyana Trofimova, Dmitry Velikanov, Artem Kuklin

**Affiliations:** 1School of Non-Ferrous Metals and Material Science, Siberian Federal University, 660041 Krasnoyarsk, Russia; hahanka@yandex.ru (A.P.); denikarp@mail.ru (D.K.);; 2Institute of Chemistry and Chemical Technology, Federal Research Center “Krasnoyarsk Science Center of the Siberian Branch of the Russian Academy of Sciences”, Akademgorodok, 660036 Krasnoyarsk, Russia; 3Kirensky Institute of Physics, Federal Research Center “Krasnoyarsk Science Center of the Siberian Branch of the Russian Academy of Sciences”, Akademgorodok, 660036 Krasnoyarsk, Russia; 4Department of Physics and Astronomy, Uppsala University, P.O. Box 516, SE-751 20 Uppsala, Sweden

**Keywords:** copper ferrite, magnetic properties, anion-exchange resin precipitation, magnetic nanoparticles

## Abstract

Copper ferrite is of great interest to researchers as a material with unique magnetic, optical, catalytic, and structural properties. In particular, the magnetic properties of this material are structurally sensitive and can be tuned by changing the distribution of Cu and Fe cations in octahedral and tetrahedral positions by controlling the synthesis parameters. In this study, we propose a new, simple, and convenient method for the synthesis of copper ferrite nanoparticles using a strongly basic anion-exchange resin in the OH form. The effect and possible mechanism of polysaccharide addition on the elemental composition, yield, and particle size of CuFe_2_O_4_ are investigated and discussed. It is shown that anion-exchange resin precipitation leads to a mixture of unstable cubic (c-CuFe_2_O_4_) phases at standard temperature and stable tetragonal (t-CuFe_2_O_4_) phases. The effect of reaction conditions on the stability of c-CuFe_2_O_4_ is studied by temperature-dependent XRD measurements and discussed in terms of cation distribution, cooperative Jahn–Teller distortion, and Cu^2+^ and oxygen vacancies in the copper ferrite lattice. The observed differences in the values of the saturation magnetization and coercivity of the prepared samples are explained in terms of variations in the particle size and structural properties of copper ferrite.

## 1. Introduction

Due to their high electrical resistivity and excellent magnetic properties, spinel ferrites are excellent candidates for modern technological applications. Copper ferrite nanoparticles (NPs) are used in biomedicine (drug delivery [1], magnetic resonance imaging [2], magnetic cell separation, and DNA extraction [3]). They also have various technological applications such as energy storage devices [4], magnetic storage media [5], and spintronic and electromagnetic devices [6,7]. Furthermore, ferrites have been used as catalysts for the photocatalytic degradation of organic matter [8,9,10], oxidation of dimethyl ether [11] and mercury [12], and reduction of 4-nitrophenol [13]. The spinel structure of ferrites provides additional sites for the catalytic reaction, leading to an increase in the efficiency of photocatalytic decomposition [14,15].

Spinels have the general formula [A^2+^B^3+^_2_O^2−^_4_], where A and B are divalent and trivalent metal cations. Copper ferrite, CuFe_2_O_4_, can be described as a cubic close-packed arrangement of oxygen ions with Cu^2+^ and Fe^3+^ ions at tetrahedral and octahedral oxygen coordination sites (A and B, respectively), so the resulting local symmetries of the two sites are different. Depending on the cation distribution in the lattice, spinels can be divided into normal and inverse types. In a normal spinel, the tetrahedral sites are occupied by A cations and the octahedral sites by B cations. In an inverse spinel, tetrahedral sites are occupied by half of the B while all A occupy octahedral sites [16,17,18].

The main method for the industrial production of ferrites remains solid-phase technology, which involves a multistep careful homogenization of the initial oxides and long-term heat treatment at high temperatures [19]. The sol-gel process proposed in recent years is time-consuming and does not always produce monophase products [20]. In addition, the resulting particles tend to fuse into large (submicron) particles [21]. Chemical precipitation is the simplest technique; however, the precipitates often trap ions and particles during formation. As the presence of impurities adversely affects the properties of the resulting materials, they must be carefully removed, making the process more complex and costly [22]. 

In this paper, we present a new approach to the production of nano-sized powders of copper ferrite anion-exchange resin precipitation [23]. This technique involves anion exchange between the resin in OH form and the solution and precipitation of hydroxides from the solution. Precipitation takes place at room temperature and ambient pressure under stable conditions at a constant pH and can be easily controlled. The particles obtained are homogeneous in composition, size, and morphology. The particles are also free of impurities and therefore do not require repeated washing and cleaning processes [24,25,26]. This technique eliminates the need for expensive equipment, provides high product yields and ensures low costs and time and energy savings. Polysaccharides with different molar masses and chain structures (dextran-40, dextran-70, and inulin) were used to tune the growth and size of the particles and to optimize the stability of the NPs. An important parameter determining the use of copper ferrite is its crystal structure. It is well known that copper ferrite can exist in two different structures: tetragonal (t-CuFe_2_O_4_) and cubic (c-CuFe_2_O_4_). t-CuFe_2_O_4_ shows better catalytic activity than c-CuFe_2_O_4_, while c-CuFe_2_O_4_ has a larger magnetic moment than the tetragonal one due to the increased concentration of Cu^2+^ at tetrahedral sites [27,28,29,30]. t-CuFe_2_O_4_ (space group I41/amd), which is stable at room temperature, forms as a result of the distortion of the cubic lattice of the bulk material under normal conditions. The cubic lattice of c-CuFe_2_O_4_ stretches along a crystallographic structure, resulting in a change in the geometry of the unit cell from cubic to tetrahedral [31]. In contrast, the undistorted cubic structure of copper ferrite (c-CuFe_2_O_4_, space group Fd-3m) exists at elevated temperatures—above 440 °C [32]. The unusual behavior of copper ferrite is explained by the d^9^ configuration of the Cu^2+^ ions, which leads to the removal of the degeneracy of the e_g_ orbitals (dz^2^ and d(x^2^ − y^2^)) and the manifestation of the cooperative Jahn–Teller effect which reduces the overall symmetry of the system [33]. 

According to Ref. [34], the coexistence of both phases is only possible in the temperature range of 360–400 °C. In contrast, Yadav et al. [35] revised this statement and emphasized the stability of the cubic phase and the coexistence of c-CuFe_2_O_4_ and t-CuFe_2_O_4_, obtained by the sol-gel method at room temperature. A number of other reports state that the structural transformation c-CuFe_2_O_4_→t-CuFe_2_O_4_ is determined by the synthesis conditions. However, despite a considerable amount of research, there is no clear correlation between the technique used and the structure of the copper ferrite. In [21], it is shown that using the sol-gel method produces tetragonal copper ferrite, while the alkaline coprecipitation produces c-CuFe_2_O_4_. In contrast, in [36] a cubic modification of copper ferrite was obtained using the sol-gel method. Furthermore, to the best of our knowledge, there is no clear explanation for why the cubic structure can remain stable at room temperature instead of transition into the tetragonal phase.

This research aims to explain the effect of polysaccharide additions on the size and structure of copper ferrite particles obtained by the anion-exchange resin precipitation method and to shed light on the parameters of the structural transition c-CuFe_2_O_4_→t-CuFe_2_O_4_ and the reasons for the stabilization of the cubic phase at room temperature.

## 2. Materials and Methods

### 2.1. Chemicals

Copper chloride (CuCl_2_·2H_2_O), iron chloride (FeCl_3_·6H_2_O), dextran ((C_6_H_10_O_5_)n Mr ~40000 Da, ~70000 Da), inulin ((C_6_H_10_O_5_)n Mr ~5000 Da), and other chemicals were of analytical grade, were purchased from Sigma-Aldrich, and were used as received. The strong-base anion-exchange resin AV-17-8 was produced by “Azot” Corporation (Cherkassy, Ukraine) in the chloride form (AV-17-8(Cl)) with a bead size of 0.4–0.6 mm (Russian GOST 20301-74). This resin is an analog of Purolite A400/A300, Lewatit M-500, Amberlite IRA 402/420, and Dowex SBR-P/Maraton A; it has a gel matrix, based on polystyrene cross-linked with divinylbenzene and the functional group quaternary ammonium (type I). The resin was washed from the monomers by treating it with 1M NaCl solution (volume ratio of resin:liquid = 1:3) for 1 h and then with 2 M NaOH (volume ratio of resin:liquid = 1:3) 6 more times for 1 h each to convert it to the OH form (AV-17-8(OH)). The resin was then thoroughly washed with water, dried at a temperature of 60 °C, and sieved. The fraction with a bead size of >0.5 mm was used in the experiments.

### 2.2. Synthesis of Copper Ferrite Nanoparticles

In typical experiments, 0.68 g CuCl_2_ · 2H_2_O and 2.16 g FeCl_3_ · 6H_2_O were dissolved in 50 mL distilled water or a polysaccharide solution (dextran or inulin) of a given concentration (dextran-40—10 weight %, dextran-70—6%, inulin—30%). AV-17-8(OH) was added in excess (150%) to the solution containing copper and ferric salts. The mixture was heated to 60° C and stirred at 180 rpm for 1 h. A sieve with round holes (0.16 mm diameter) was used to remove the anion-exchange resin beads; the precipitate was centrifuged, washed with distilled water, dried in air at 80 °C, and then heated in a muffle furnace at 800 or 900 °C for 1 h.

The resin was also washed with distilled water and then treated three times with 1 M HNO_3_ (10 mL portions) while stirring for 1 h. All liquid samples (eluates, stock solutions, dissolved precipitates) were analyzed for copper and iron ions using a Perkin Elmer A Analyst 400 Atomic Absorption spectrometer (USA). The product percent yield (η) and mole fraction of metals in the resin (χ) were defined as the ratio of the total amount of copper and iron moles in the product or eluate to the total amount of the copper and iron moles in the stock solution.

The rate of anion-exchange resin precipitation was studied by measuring the ionic conductivities (λ) of reaction solutions on a Multitest KSL-101 (Semico, Novosibirsk, Russia) conductance meter. 

### 2.3. Nanoparticle Characterization

X-ray powder diffraction data were obtained using an X’Pert PRO diffractometer (PANalytical) with CuKα radiation. PIXcel (PANalytical), equipped with a graphite monochromator, was used as a detector. The sample was ground in an agate mortar and prepared by the direct cuvette loading method. Scanning conditions: ranged from 3 to 100° on the 2θ scale with a step size of 0.013°, ∆t–50 sec/step, in air at room temperature. High-temperature X-ray studies were carried out by using an HTK1200N (Anton Paar) high-temperature chamber in the air. To prepare the sample, about 0.2 g of the substance was loaded into an alumina cuvette. The sample was heated to a certain temperature at a constant rate of 50 °C/min and then scanned within 5 min. Phase identification was carried out using the PDF-2 database card file. Rietveld refinement of compositions and cell parameter extraction of obtained materials were carried out in Topas (version 7) software. 

TEM analysis was carried out using a Hitachi 7700M (Hitachi Corporation, Hitachi, Japan, accelerating voltage: 110 kV). A copper ferrite particle size distribution histogram was obtained from more than 300 particles. The FTIR spectra of samples were recorded on a Tensor 27 (Bruker, Germany) FTIR spectrometer in the range of 4000–400 cm^−1^. 

The magnetic properties of the obtained material were investigated in a magnetic field up to ±15 kOe at 298 K using a vibrating sample magnetometer. An electromagnet with high magnetic field uniformity was used as a source. The magnetic measurements were performed using a direct method of measuring the inductive electromotive force. The mechanical vibrations of the sample were provided by a vibrator of the original design. The relative instability of the oscillation amplitude was 0.01%, with a frequency of 0.001%. The registration of the signal was conducted using a system of four pickup coils. The dynamic range of the device was 5·10^−6^–10^2^ emu.

## 3. Results and Discussion

### 3.1. The Effects of Polysaccharides with Different Molar Masses on the Anion-Exchange Resin Precipitation Process

Copper ferrite nanoparticles were synthesized by an anion-exchange resin precipitation process, which can be described by the equation:FeCl_3_ + CuCl_2_ + 5R-OH →Fe(OH)_3_ + Cu(OH)_2_ + 5R-Cl,(1)

R-OH and R-Cl are the anion-exchange resin AV-17-8 in the OH form and Cl form. 

Anion-exchange resin precipitation involves two interconnected chemical reactions. In the first reaction, anions from a solution are exchanged for anions attached to an immobile solid particle: Cl^−^ + R-OH → OH^−^ + R-Cl. (2)

The second process is the deposition of metals as insoluble hydroxides:Fe^3+^ + Cu^2+^ + 5 OH^−^→Fe(OH)_3_ + Cu(OH)_2_.(3)

As the copper and iron cations are bound in their hydroxides and Cl anions transfer into the sorbent phase, the equilibrium shifts in the direction of the products, and the ionic strength and the ionic conductivity of the solution decrease. Reducing the ionic strength of the solution facilitates the production of particles with identical composition and size and without impurity ions. The resulting materials have a large surface area and high reactivity [23,24,25,26,37]. 

The deposition of metal hydroxides starts at the resin–solution interface, namely on the resin beads. When the thickness of the surface deposit reaches 1–1.5 μm, it flakes off and a single product phase is formed [26,34]. The size of the produced particles can be controlled by reaction parameters and by the use of stabilizers to prevent agglomeration and aggregation [24]. 

In this study, polyglucans and polyfructans with a molar mass of 5000 to 70,000 Da (inulin, dextran-40, and dextran-70) were used to disperse the nanoparticles and keep them in a stable colloidal state. Since the solution viscosity affects the mass transfer and mass exchange, we used polysaccharide solutions with the same kinematic viscosity of 5 × 10^−6^ m^2^/s, which does not complicate the resin separation, so the concentrations of the solutions were different for different polysaccharides (dextran-40—10 wt.%, dextran-70—6 wt.%, inulin—30 wt.%). 

A study of the rate of the anion-exchange resin precipitation gave the data shown in Figure 1. When dextran-40 and inulin were added, or in the absence of any polysaccharides, precipitation proceeded quite rapidly: the specific electrical conductivity of the reaction solutions was dramatically reduced to zero within 10 min, and the extent of precipitation reached 99%. At the same time, the addition of dextran-70 drastically reduced the process rate: precipitation occurred within 60 min. In our opinion, in the presence of dextran-70, a surface deposit with higher adhesive properties was formed on the resin beads, which impeded the diffusion of ions through this layer and reduced the precipitation rate. The deposit flaked off the resin beads after 30–60 min [24,26].

The resulting anion-exchange precipitation products had an X-ray amorphous nature. The elemental composition and yield of the product as well as its phase composition after annealing depend on the type of polysaccharides (polyglucan or polyfructan) and their molar masses (Table 1). XRD (Figure 2 and Appendix A) showed the formation of copper ferrite as a single-phase structure after annealing at 800 °C for the samples obtained with dextran-40 or inulin and without the polysaccharides. All the diffraction peaks are in good agreement with the JCPDF file for CuFe_2_O_4_ (JCPDF №74-8585 and №34-0425). In the case of the sample with dextran-70, the product contained hematite impurities due to the non-stoichiometry of the precursor. The product yield was only 80%, and the mole fraction of metals in the resin reached 9.8%. In contrast, the addition of dextran-40 resulted in a low metal content in the resin phase (2%) and a high product yield (98%). This could reflect the difference in the adhesive properties of the surface precipitates, which could be related to the adsorption of the polysaccharide molecules onto the nanoparticles. As mentioned above, the adsorption of dextran-70 on the nanoparticles prevents the exfoliation of the surface deposit. Conversely, the adsorption of dextran-40 or inulin increases the colloidal stability of the nanoparticles and leads to the peptization of the surface deposit. 

The adsorption of polysaccharides on the as-prepared samples was confirmed by IR spectroscopy (Figure 3). The absorption bands at 2853 and 2923 cm^−1^ refer to the symmetric and asymmetric vibration of the CH_2_ groups of the polysaccharides. The bands have different intensities which can be attributed to the amount of polysaccharide adsorbed on the nanoparticles. The highest adsorption is observed for the sample obtained with dextran-40 (Figure 3, curve 1). In the case of the sample obtained with dextran-70, the content of the organic phase on the surface of the particles is minimal (Figure 3, curve 3). The adsorbed polysaccharides are completely removed during annealing of the products at temperatures above 300 °C (Figure 4 and Appendix A). The IR spectrum of the powder with dextran-40 additive calcined at 800 °C for 3 h consists of only one vibration band at 590 cm^−1^ corresponding to the Fe-O stretching of the ferrite structure (Figure 3, line 5).

The polysaccharides influence the growth process of the copper ferrite particles. By adjusting the type and molar mass of the polysaccharide additives, we can modify the size of the product. Thus, according to the TEM data (Figure 5), the use of dextran with an average molar mass of ~40,000 g/mol led to the formation of copper ferrite nanoparticles with a size of 14 ± 3 nm; in the presence of dextran-70, particles of 87 ± 24 nm were obtained; particles of 63 ± 14 nm were synthesized with inulin; and particles of 134 nm were produced without any polysaccharides.

The rate of particle aggregation is an important factor in controlling the size of the crystallites in the final product. If the surface of the colloid adsorbs polysaccharide molecules, the growth rate of the colloids is limited, and agglomeration of the nanoparticles is avoided. The adsorption efficiency is determined by the stability of the surface complexes of the metal hydroxide with polysaccharides [38]. The binding strength of polysaccharide molecules to the colloid surface depends on the surface area available for binding, the strength of the hydrogen bonds, and the steric barriers between the polymer molecules. These factors are determined by the molar mass of the polysaccharide and the conformation of its chain: low-molar-mass polysaccharides are represented by linear chains, while spiral and globular chains are formed by medium-molar-mass polysaccharides. A decrease in the molar mass of the polysaccharide leads to a decrease in the number of bonds and hence a decrease in the stability of the surface complexes. On the other hand, globules formed from polysaccharides with high molar masses have a small space available for binding to the surface of the particle. Thus, there is an optimal polysaccharide chain length, which, as shown in Ref. [38], is approximately equal to the length of the circumference of the nanoparticle. The circumferential length of 14 nm particles (sample 2, Table 1) corresponds to the chain length of a polysaccharide with a molar mass of 40,000 Da. In this case, the most stable colloidal system is formed and particles with a minimum size are produced [24,38]. These results are similar to our previous study where CoFe_2_O_4_ nanoparticles with a size of 15 nm were obtained in the presence of dextran-40 [39].

The simplicity and time and cost efficiency due to the lack of expensive equipment make the proposed method for the synthesis of copper ferrite nanoparticles convenient, easily reproducible, and scalable under laboratory conditions. In addition, the use of the anion-exchange resin technique makes it possible to obtain an uncontaminated product with reproducible physicochemical properties. The addition of dextran-40 results in the formation of stable colloidal dispersions. The high zeta potential of −30.6 mV indicates that the hydrosols obtained are not only sterically but also electrically stabilized. The resulting nanoparticles can be used as effective catalysts as well as magnetic nuclei in the creation of hybrid “core-shell” nanostructures, which have potential applications in biotechnology, nanomedicine, and theranostics [40,41,42].

### 3.2. Control of Structural Parameters of Copper Ferrite

#### 3.2.1. The Effect of Particle Size on c-CuFe_2_O_4_ Stability

The most important factor determining the use of copper ferrite is the structure of its crystal lattice. As mentioned above, CuFe_2_O_4_ has a high-temperature cubic modification and a low-temperature tetragonal modification. However, c-CuFe_2_O_4_ can also exist at room temperature. Several explanations for this phenomenon have been proposed in the literature. Some papers [43,44,45,46] suggest that the main factor in the stability of the cubic modification of copper ferrite is the particle size. Particles of <40 nm contribute to the cubic phase stabilization, whereas larger submicron and micron particles lead to the formation of tetragonal copper ferrite. In [47], it was also shown that the particle size has a significant effect on the temperature of the phase transformation of c-CuFe_2_O_4_→t-CuFe_2_O_4_, which decreases sharply for 15 nm nanoparticles.

Our results show that although the particle size varied from 14 nm to 134 nm, the change in cubic phase content was insignificant (22% to 33%), and there is no apparent correlation between these values. 

Our results are in agreement with those obtained in [21,46], where no correlation between the dimensional effect and the phase of the copper ferrite was observed. In [46], the sol-gel combustion method was used to obtain c-CuFe_2_O_4_ with a particle size of >100 nm. Submicron particles of stable cubic copper ferrite (Fd-3m) were synthesized under standard conditions by the coprecipitation technique [21]. In addition, the tetragonal phase of copper ferrite can also contain small particles. For example, t-CuFe_2_O_4_ with a particle size of 15-25 nm has been reported [48,49]. 

#### 3.2.2. The Effect of the Annealing Temperature on c-CuFe_2_O_4_ Stability

Copper ferrite is known to be an inverse spinel characterized by the presence of copper ions only in octahedral positions and iron in tetrahedral and octahedral positions. However, some copper ions occupy tetrahedral sites as a result of cationic migration of Cu^2+^ and Fe^3+^. Thus, a so-called “mixed” spinel is formed, containing Cu and Fe in both tetrahedral (A) and octahedral (B) positions. The resulting distribution of Cu^2+^ and Fe^3+^ ions in such a structure can be represented as follows: (Cu_x_Fe_1-x_)^A^[Cu_1-x_Fe_1+x_]^B^O_4_, where x is an inversion parameter; if x = 0, the spinel is inverted, and if x = 1 the spinel is normal. A high concentration of Cu^2+^ ions per formula unit in octahedral positions (1 − x > 0.8) leads to a tetragonal distortion of the cubic spinel due to the cooperative Jahn–Teller effect [50,51,52].

It has been shown in [35,53] that the distribution of cations between tetrahedral and octahedral positions, which is responsible for the evolution of structural phases, increases with increasing annealing temperature. Furthermore, the non-equilibrium distribution of Cu atoms can be maintained at room temperature. In contrast, [29,54] show that the high annealing temperature contributes to the formation of the tetragonal CuFe_2_O_4_, and the structural change c-CuFe_2_O_4_ →t-CuFe_2_O_4_ occurs as the annealing temperature increases. In [45], cubic copper ferrite was obtained at a temperature below 300 °C, while a tetragonal phase was produced above 400 °C. It is also mentioned that the formation of cubic CuFe_2_O_4_ spinel films is independent of their calcination temperature [55].

The obtained products were annealed at 800 and 900 °C for 1 h (Table 2 and Table 3). According to XRD (Figure 6 and Appendix A), a mixture of cubic and tetragonal modifications of copper ferrite with a different molar ratio of c-CuFe_2_O_4_/t-CuFe_2_O_4_ is formed in all experiments. However, the relative content of the cubic phase increases with increasing temperature in samples 2 and 4 and does not change in samples 1 and 3.

A detailed study of the evolution of the crystal structure of copper ferrite as a function of temperature was carried out using high-temperature X-ray diffraction. The in situ experiments were carried out on an X’Pert PRO diffractometer (PANalytical) (see Section 2.3) equipped with an HTK1200N high-temperature chamber (Anton Paar). Two samples without polysaccharides and with dextran-40 (1 and 2, Table 1) were analyzed. Heating was carried out from 25 °C to 900 °C at a rate of 50 °C/min. X-ray patterns were recorded every 100 °C. After 10 min of exposure at 900 °C, the sample was cooled to 25 °C. Figure 7 shows fragments of X-ray powder patterns in the angular range of 33–37° 2θ, corresponding to the reflection (311) of c-CuFe_2_O_4_ with the maximum intensity, in the temperature interval of 400–900 °C.

The samples show similar behavior when heated. X-ray amorphous powder patterns are observed up to 500 °C (Figure 7). The cubic phase starts to form at 500 °C and is fully formed at 800 °C. The intensity of the main reflection (311) of c-CuFe_2_O_4_ increases with temperature. Sample 2, prepared with dextran-40, crystallizes at a lower temperature. At 900 °C, both samples contain only the cubic phase of copper ferrite. Part of the cubic copper ferrite changes to a tetragonal modification upon cooling. Approximately 47% of the cubic phase remains in each sample (see Appendix A for Appendix A, Appendix A). The results indicate that neither the addition of the polysaccharide nor the particle size significantly affect the stability of the cubic copper ferrite modification.

#### 3.2.3. The Effect of the Cooling Rate on c-CuFe_2_O_4_ Stability

The high-temperature X-ray diffraction chamber was cooled by water at a rate of approximately 30 °C/min. Additional experiments with lower and higher cooling rates were performed on the sample obtained without polysaccharides (1, Table 1). After heating to 900 °C for 30 min with a dwell time of 60 min at 900 °C, one sample (denoted as 1f; letter ‘f’ is from ‘furnace’) remained in the furnace for 7 h until it cooled to 25 °C, and the second sample (denoted as 1q; letter ‘q’ is from ‘quenching’) was removed from the furnace and cooled under room conditions, reaching room temperature in 7 min. The temperature of the samples was monitored during the cooling time (Appendix A). The maximum cooling rate was 2.6 and 0.1 °C/s for the 1q and 1f samples, respectively, and cooling slowed down at lower temperatures.

Table 4 and Figure 8 (also Appendix A) show the phase composition of samples 1q and 1f and the samples with a non-stoichiometric molar ratio of Cu to Fe: +15% Cu (Cf and Cq); +15% Fe (Ff and Fq). The heating and cooling of the non-stoichiometric samples were performed as described above: samples Cq and Fq were rapidly cooled (quenched), while samples Cf and Ff were slowly cooled. Quenching the samples, irrespective of their chemical composition, led to a significant increase in the relative content of c-CuFe_2_O_4_. These results are in agreement with the literature data [46,50,51]. It has been shown that quenching the samples stabilizes their non-equilibrium high-temperature phase, where copper cations are distributed between tetrahedral and octahedral lattice positions with equal probability. Alternatively, furnace cooling is slow enough to establish the equilibrium distribution of Cu, resulting in tetragonal distortion [56].

#### 3.2.4. The Effect of Elemental Composition on c-CuFe_2_O_4_ Stability

Most of the investigated samples, regardless of their n(Cu)/n(Fe) molar ratio, contained monoclinic CuO as a second phase (Table 4), even in the case of a 15% excess of iron (sample Fq). At the same time, no separate iron oxide phase was detected (Figure 8). The amount of CuO correlates with the relative content of cubic copper ferrite, in good agreement with known results [21,57,58]. We suggest that CuO is formed during the cooling and stabilization of the cubic modification. Appendix A and Appendix A show the phase composition of samples 1 and 2 (Table 1), determined at 900 °C during high-temperature X-ray diffraction measurements and after cooling to 25 °C. c-CuFe_2_O_4_ is the only phase observed at high temperatures. Upon cooling, the peaks of the cubic phase are reduced, and a doublet corresponding to the tetragonal phase appears along with the peaks corresponding to CuO. It is noted that the formation of CuO occurs during the thermal decomposition of CuFe_2_O_4-δ_ in the temperature range of 900–1100 °C [59]:CuFe_2_O_4-δ_ = 4δCu_0.5_Fe_2.5_O_4_ + (1-5δ)CuFe_2_O_4_+ 3δCuO(4)

The composition CuFe_2_O_4-δ_ corresponds to the non-stoichiometric oxygen-deficient copper ferrite formed by the loss of oxygen during the high-temperature heat treatment of the precursors [51]. Oxygen deficiency is represented by oxygen vacancies and the value of δ varies with the partial pressure of oxygen and the heating temperature [60]. The loss of oxygen is followed by a partial reduction of Cu^2+^ to Cu^+^ [35]. As the ferrite cools, secondary oxidation leads to the segregation of tenorite CuO and the formation of copper ferrite with copper deficiency and iron excess—Cu_1-η_Fe_2+η_O_4_, where η varies in a wide range from 0.04 to 0.5. These deviations from stoichiometry are typical for cubic copper ferrite [59,60,61]. We suggest that the formation of copper ferrite with excess (superstoichiometric) iron is the reason for the absence of a separate phase of iron oxides in samples Ff and Fq. These compositional changes during annealing initiate the transition of Cu^2+^ from octahedral to tetrahedral positions. Thus, the stabilization of the cubic phase is not only caused by the migration of Cu^2+^ but is also related to the change in oxygen content. The oxygen diffusion depends on the duration of the high-temperature treatment of copper ferrite, which is not considered in many studies and may be the cause of the controversial results reported previously.

It should be noted that the loss of oxygen during heating may be irreversible if the samples are quenched or reversible at a low cooling rate. Quenching of CuFe_2_O_4_ stabilizes as Cu^2+^ at tetrahedral sites as well as oxygen and copper vacancies in the crystal lattice. These processes are responsible for the stability of the non-stoichiometric phase of cubic copper ferrite at room temperature. The excess of iron ions in the precursor also contributes to this effect. The synergy of these factors is illustrated by the Fq sample (Table 4) where the maximum content of c-CuFe_2_O_4_ is observed.

### 3.3. The Effect of the CuFe_2_O_4_ Nanoparticles’ Structure on Their Magnetic Properties

Figure 9 shows the dependence of the magnetization of the CuFe_2_O_4_ nanoparticles obtained without polysaccharides and with dextran-40 (samples 1 and 2, Table 1) on the applied magnetic field measured at 298 K. The values of the saturation magnetization (Ms) were estimated in magnetic field H = ±15 kOe. The specific residual magnetizations (Mr) were determined from the intersection of the falling part of the magnetization curve with the Y axis. The values of the coercivity (Hc) of the nanoparticles were determined by measuring the hysteresis loop width. The basic parameters of the hysteresis loops are summarized in Table 5.

The observed saturation magnetization (Ms) value of sample 2 is slightly higher than that of sample 1 due to the larger amount of the cubic phase. Both samples show typical ferrimagnetic behavior. Since the magnetic moments of copper (II) ions and iron (III) ions are very different, the magnetic properties of CuFe_2_O_4_ depend strongly on the distribution of cations in octahedral and tetrahedral positions [40]. The total magnetic moment (µ) in the ferrimagnetic configurations (Cu_x_Fe_1-x_)^A^[Cu_1-x_Fe_1+x_]^B^O_4_ can be calculated by a formula:µ = µCu + 2x(µFe − µCu),(5)
where µFe and µCu are the magnetic moments of ions Fe^3+^ and Cu^2+.^ This formula shows that the total magnetic moment increases when the amount of copper ions in tetrahedral positions increases, which is typical for c-CuFe_2_O_4_.

The observed Ms values of samples 1 and 2 are close to each other and significantly lower than the bulk value of t-CuFe_2_O_4_ (74.1 emu/g) [62,63]. It is known that the value of MS decreases with decreasing crystallite size. Such behavior of the nanocrystalline samples could be related to the large surface area of the nanoparticles leading to significant adsorption of impurity atoms on the surface and structural disorder of the surface atoms [44]. As the particle size of sample 2 is much smaller (14 ± 3 nm) than that of sample 1 (134 ± 23), its saturation magnetization should be significantly lower. However, as the particle size decreases, the c-CuFe_2_O_4_/t-CuFe_2_O_4_ ratio increases, leading to an increase in the net saturation magnetization value. Thus, the size effect and the cationic disordering effect act in opposite directions, resulting in relatively similar Ms values for both samples.

In contrast, the coercivity values (Hc) of the samples differ significantly. The coercivity of magnetic nanomaterials is very sensitive to size variations and decreases to zero with decreasing particle size when the magnetic multidomain state changes to a single-domain state and the particle enters the superparamagnetic regime [64]. The ferromagnetic-to-superparamagnetic transition in CuFe_2_O_4_ nanoparticles is not only due to the size effect. The coercivity increases with the structural transformation c-CuFe_2_O_4_→t-CuFe_2_O_4_ due to the large anisotropy resulting from the tetrahedral distortion [65].

Thus, the magnetic properties of the copper ferrite nanoparticles depend on their structural characteristics and can be tuned by varying the reaction parameters (the molar ratio of the reactants used in the reaction, the temperature and time of the heat treatment process, the cooling rate, etc.). These conditions vary from paper to paper, which may explain the diversity of the obtained results. Some of these conditions are often not controlled properly, which may explain why the experimental results obtained in different research papers do not correlate with each other. At the same time, the magnetic parameters of the copper ferrite nanoparticles measured here are in agreement with those reported elsewhere: Ms = 31.4 emu/g, Hc = 400.3 Oe [20]; Ms = 27.4 emu/g, Hc = 526.6 Oe [21]; Ms = 41.1 emu/g, Hc = 241 Oe [66]; and Ms = 32.4 emu/g, Hc = 517 Oe [67].

## 4. Conclusions

In this paper, a new cost-, time-, and energy-efficient technique using anion-exchange resin in OH form as a reaction agent has been proposed to obtain homogeneous copper ferrite nanoparticles with uniform size and morphology without any impurity ions. The technique does not require expensive equipment and ensures a high product yield. The addition of polysaccharides with different molar masses and chain structures during precipitation significantly influences the growth and agglomeration of the particles. The use of inulin with an average molar mass of 5 kDa led to the formation of particles with a size of 63 ± 14 nm; in the presence of dextran (70 kDa), particles with a size of 87 ± 24 nm were obtained; the use of dextran (40 kDa) resulted in particles with a size of 14 ± 3 nm; and the absence of polysaccharide addition led to particles with a size of 134 ± 23 nm. Our high-temperature XRD study has shown that a significant part of the cubic copper ferrite formed at 500 °C remains stable at room temperature. We examined possible reasons for the stability of c-CuFe_2_O_4_ unstable at normal conditions and revealed that polysaccharide additives, particle size, and annealing temperatures in the range of 800–950 °C do not significantly affect the cubic phase stability. We have shown that the relative content of the cubic phase in the samples increases significantly when the samples are rapidly cooled in air (quenched) and when there is an excess of iron ions in the precursors. These factors contribute to the formation of oxygen and Cu^2+^ vacancies in the ferrite crystal lattice and cation distribution, resulting in the stabilization of the cubic copper ferrite at normal conditions. Despite the structural sensitivity of the magnetic properties of CuFe_2_O_4_, we did not find significant differences in the values of saturation and residual magnetization for the samples with different contents of cubic phases due to the large differences in particle size in the samples (between 14 and 134 nm). However, the coercivity values of the samples are almost twice as different due to the size effect and the magnetic anisotropy caused by the tetragonal distortion. We believe that it is possible to tune the magnetic properties of copper ferrite by controlling the reaction parameters for its prospective use in various areas such as magnetic, microwave, and biomedical applications.

## Figures and Tables

**Figure 1 materials-16-02318-f001:**
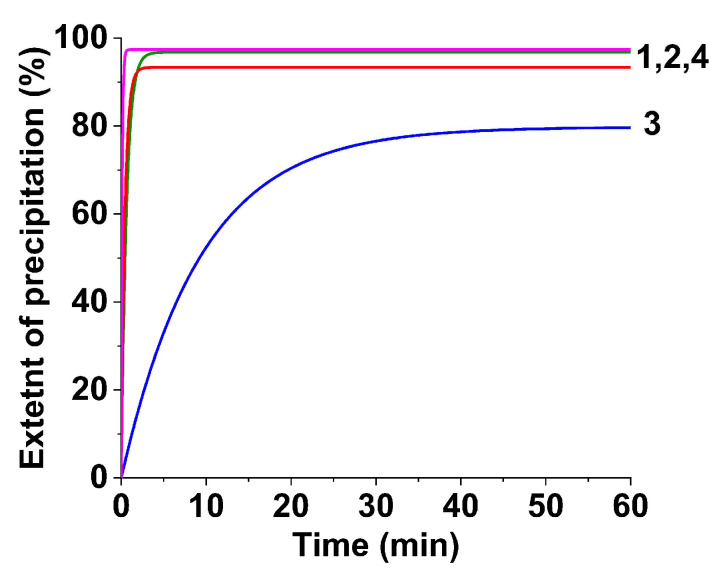
The extent of the precipitation of metals as a function of time: 1—obtained with dextran-40, 2—obtained with inulin, 3—obtained with dextran-70, 4—obtained without polysaccharides.

**Figure 2 materials-16-02318-f002:**
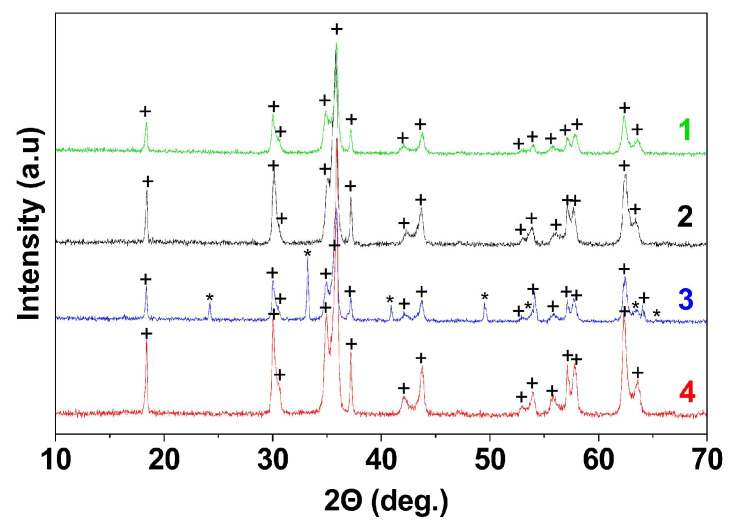
X-ray diffraction patterns of CuFe_2_O_4_ powders annealed at 800 °C: 1—sample obtained without polysaccharides, 2—obtained with dextran-40, 3—obtained with dextran-70, 4—obtained with inulin, + CuFe_2_O_4_, * Fe_2_O_3_.

**Figure 3 materials-16-02318-f003:**
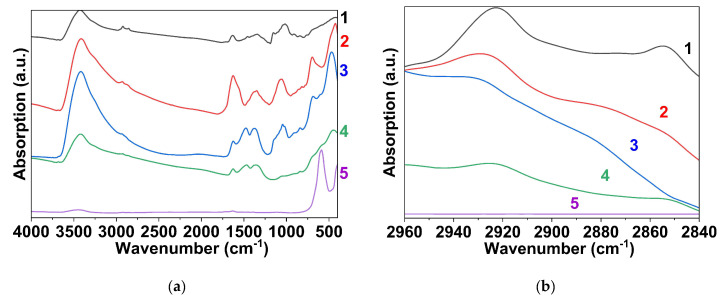
FTIR spectra in the spectral regions of 4000−500 cm^−1^ (**a**) and 2960−2840 cm^−1^ (**b**) of the as-prepared samples obtained using dextran-40 (1), inulin (2), dextran-70 (3), and without polysaccharides (4) and of sample 2 (Table 1) with dextran-40 additive calcined at 800 °C (5).

**Figure 4 materials-16-02318-f004:**
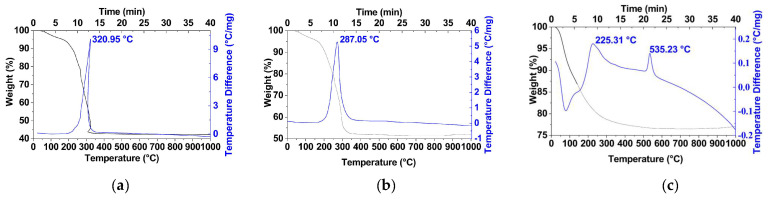
TGA (black curve) and DSC (blue curve) curves for the as-prepared samples with dextran-40 (**a**) and inulin (**b**) and without polysaccharides (**c**). The maximums on the DSC curves correspond to complete oxidation of the adsorbed polysaccharide.

**Figure 5 materials-16-02318-f005:**
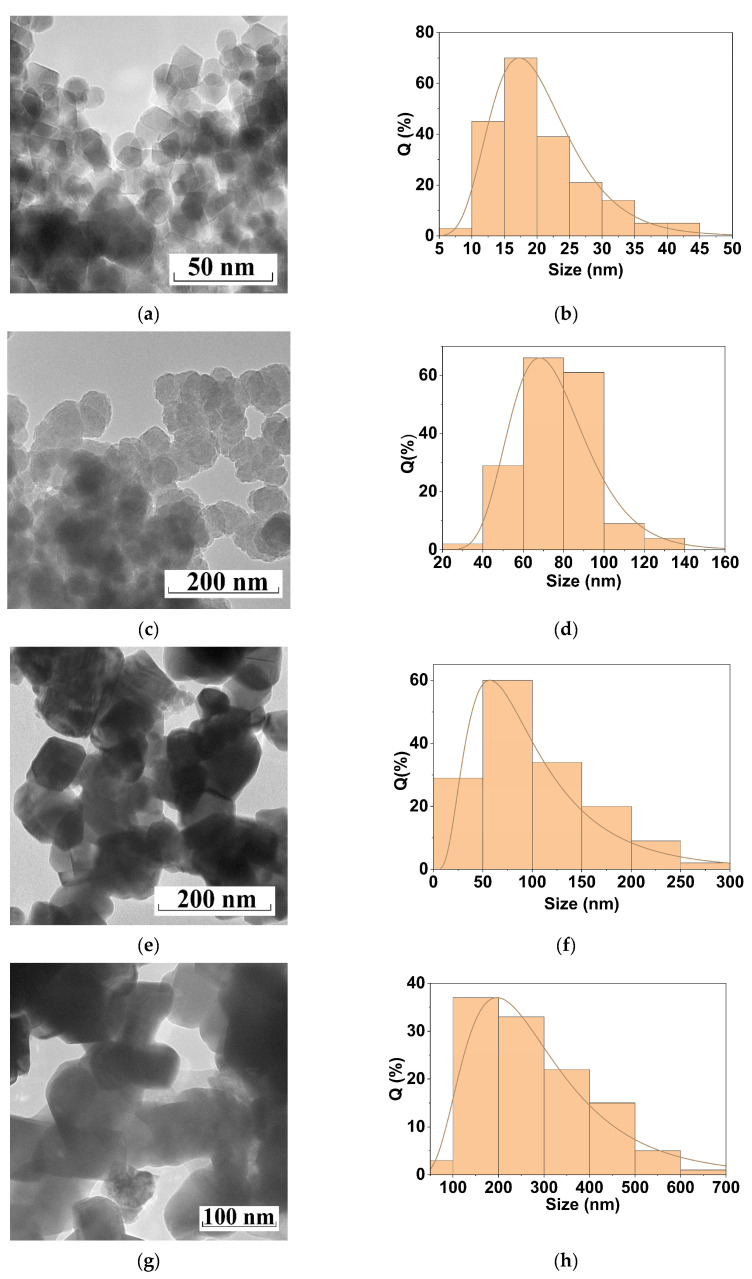
TEM images (**left** panel) and the particle size distribution diagrams (**right** panel) for the samples of CuFe_2_O_4_ calcined at 800 °C: (**a**,**b**) obtained with dextran-40; (**c**,**d**) obtained with inulin; (**e**,**f**) obtained with dextran-70; and (**g**,**h**) obtained without polysaccharides.

**Figure 6 materials-16-02318-f006:**
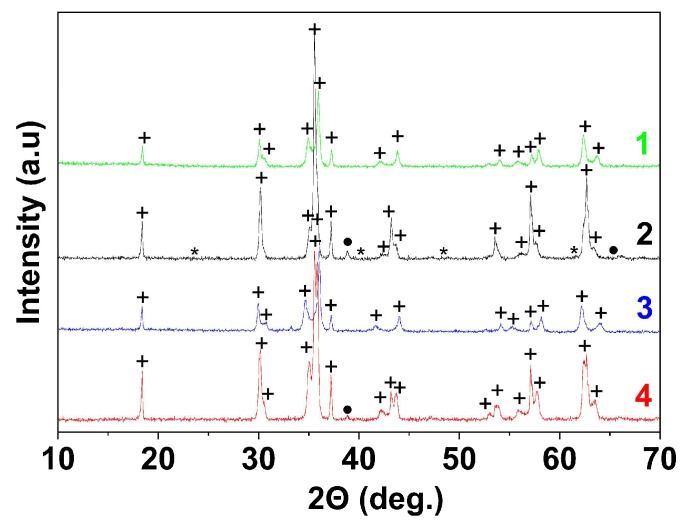
X-ray diffraction patterns of CuFe_2_O_4_ powders annealed at 900 °C: 1—sample obtained without polysaccharides, 2—obtained with dextran-40, 3—obtained with dextran-70, 4—obtained with inulin, + CuFe_2_O_4_, * Fe_2_O_3_, • CuO.

**Figure 7 materials-16-02318-f007:**
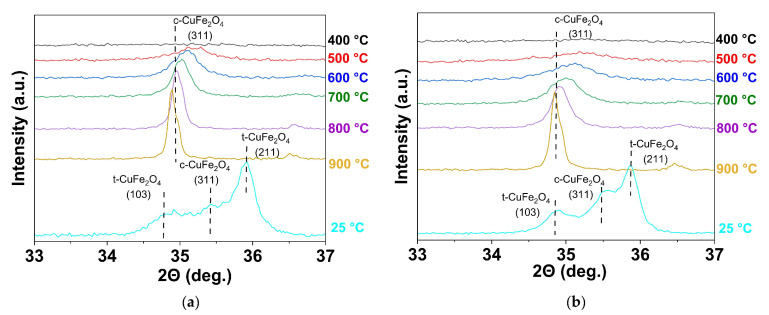
Magnified region of the X-ray diffraction pattern of the as-prepared samples 1 and 2 (**a**) obtained without polysaccharides; (**b**) obtained with dextran-40) in the temperature range of 400–900 °C with the angular spacing corresponding to the main reflection (311) of the cubic CuFe_2_O_4_.

**Figure 8 materials-16-02318-f008:**
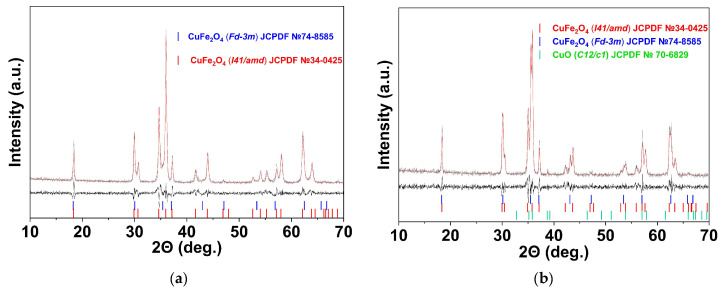
Phase composition of samples Ff (**a**) and Fq (**b**). The red line indicates the calculated model. The difference between both values and the line diagram of phases (c-CuFe_2_O_4_, t-CuFe_2_O_4_, CuO) are presented in the lower portion of the graph are indicated by the black line.

**Figure 9 materials-16-02318-f009:**
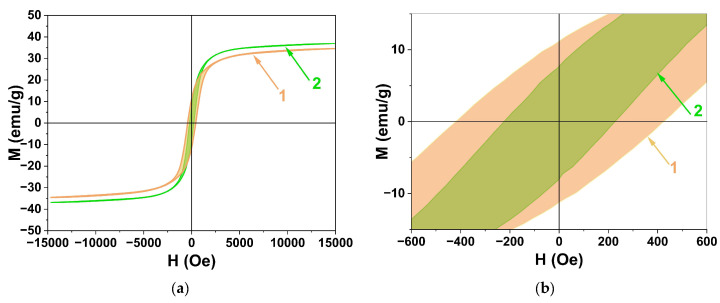
The magnetization curves (**a**) and their partial enlarged detail (**b**) measured in magnetic field H = ±15 kOe at 298 K: 1—without the use of polysaccharides and 2—using dextran-40.

**Table 1 materials-16-02318-t001:** Effect of polysaccharides on copper ferrite synthesis and nanoparticle size.

Sample	Polysaccharide	The Mole Fraction of Metals in the Resin, %	The Mole Ratio of Cu to Fe in the Product (n_Cu_/n_Fe_)	Product Yield, %	Average Size of Nanoparticles (TEM), nm	Phases after Annealing
1	-	3.0	0.5 ^1^	96.0 ± 0.6	134 ± 23	CuFe_2_O_4_
2	Dextran-40	2.0	0.5	98.0 ± 0.6	14 ± 3	CuFe_2_O_4_
3	Dextran-70	9.8	0.4	80.0 ± 1.0	87 ± 24	CuFe_2_O_4_, Fe_2_O_3_
4	Inulin	2.5	0.5	97.0 ± 0.6	63 ± 14	CuFe_2_O_4_

^1^ Mole ratio of copper to ferrite 0.5 is stoichiometric for copper ferrite CuFe_2_O_4_.

**Table 2 materials-16-02318-t002:** Cell parameters and phase composition of the samples obtained by ion-exchange resin precipitation and annealed at 800 °C for 1 h.

Sample	Polysaccharide	t-CuFe_2_O_4_ (I41/amd)			c-CuFe_2_O_4_ (Fd-3m)		Fe_2_O_3_	χ^2^
		a	c	ω, wt%	a	ω, wt%	ω, wt%	
1	-	5.853 ± 0.001	8.591 ± 0.001	76 ± 2	8.391 ± 0.001	24 ± 2	-	1.237
2	Dextran-40	5.870 ± 0.001	8.556 ± 0.001	67 ± 2	8.388 ± 0.001	33 ± 2	-	1.413
3	Dextran-70	5.860 ± 0.001	8.581 ± 0.001	56 ± 2	8.385 ± 0.001	22 ± 2	22 ± 2	1.266
4	Inulin	5.857 ± 0.001	8.585 ± 0.001	75 ± 2	8.385 ± 0.001	25 ± 2	-	1.582

**Table 3 materials-16-02318-t003:** Cell parameters and phase composition of the samples obtained by ion-exchange resin precipitation and annealed at 900 °C for 1 h.

Sample	Polysaccharide	CuFe_2_O_4_ (I41/amd)			CuFe_2_O_4_ (Fd-3m)		Fe_2_O_3_	CuO	χ^2^
		a	c	ω, wt%	a	ω, wt%	ω, wt%	ω, wt%	
1	-	5.851 ± 0.001	8.596 ± 0.001	78 ± 3	8.390 ± 0.001	22 ± 2	-	-	1.322
2	Dextran-40	5.876 ± 0.001	8.545 ± 0.001	33.6 ± 0.9	8.387 ± 0.001	62.7 ± 0.9	2.3 ± 0.7	1.4 ± 0.3	1.232
3	Dextran-70	5.824 ± 0.001	8.673 ± 0.001	79 ± 1	8.385 ± 0.001	21 ± 1	-	-	1.391
4	Inulin	5.867 ± 0.001	8.558 ± 0.001	60.0 ± 0.8	8.384 ± 0.001	39.4 ± 0.8	-	0.6 ± 0.2	1.594

**Table 4 materials-16-02318-t004:** Phase composition of the samples with different mole ratios of Cu to Fe annealed at 900 °C and cooled at various rates.

Sample	Mole Ratio n(Cu)/n(Fe)	Cooling Mode	ω(t-CuFe_2_O_4_), %	ω(c-CuFe_2_O_4_), %	ω(CuO), %	χ^2^
1f	0.5	In furnace	81 ± 2	16 ± 2	3.0 ± 0.2	1.449
1q		Quenching	67 ± 1	29 ± 1	4.0 ± 0.3	1.358
Cf	0.6	In furnace	79 ± 2	14 ± 2	7.0 ± 0.4	1.439
Cq		Quenching	68 ± 1	23 ± 1	9.0 ± 0.4	1.339
Ff	0.4	In furnace	84 ± 2	16 ± 1	-	1.506
Fq		Quenching	55 ± 1	44 ± 1	1.0 ± 0.3	1.289

**Table 5 materials-16-02318-t005:** Magnetic parameters obtained from the results of magnetic hysteresis loops in Figure 9.

Sample	Polysaccharide	c-CuFe_2_O_4_/ t-CuFe_2_O	Ms, emu/g	Mr, emu/g	Hc, Oe	Size of Nanoparticles (TEM), nm
1	-	0.3	34.6	10.6	417.0	134 ± 23
2	Dextran-40	0.6	36.9	6.8	220.0	14 ± 3

## Data Availability

Not applicable.

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
