# Peer review of "Copper Ferrite Nanoparticles Synthesized Using Anion-Exchange Resin: Influence of Synthesis Parameters on the Cubic Phase Stability"

_materials, 2023, doi:10.3390/ma16062318_

Round 1

Reviewer 1 Report

In this manuscript, the authors reported the synthesis of copper ferrite nanoparticles using anion exchange resin. Polyglucan and polyfructan with a molar mass from 5000 to 70000 Da were used to disperse nanoparticles, and their effect on the product yield and nanoparticle size was studied. In addition, the influence of nanoparticle size, annealing temperature, cooling rate, and elemental composition on the cubic phase stability was also studied. they also studied the effect of the CuFe2O4 nanoparticles structure on their magnetic properties. It is certain that the report would attract researchers in related research fields. Thus, I suggest the acceptance of the manuscript after solving these major problems.

1. It is hard to recognize the lines in Figure 1, the authors should use different colors to distinguish the different lines.

2. In Figures 2 and 6, the PXRD pattern of CuFe2O4 should also be presented, as in Figure 8. Then it would be easier to compare the matching degree of the products with CuFe2O4.

3. It is clear that the absorption bands at 2853 and 2923 cm–1 of the sample obtained with dextran-70 are not negligible. The peaks can be clearly seen in Figure 3a, please revise it.

4. All the figures should be improved, including the definition and aesthetic.

5. In 3.2.1, the cubic phase contents are not only affected by the particle size but also the polysaccharide. It is better to use the same polysaccharide to synthesize particles of different sizes and then compare their cubic phase contents. 

6. The authors mentioned many times about supporting information, but it can not be found in the submitted files.

Author Response

Reply: We are grateful to the reviewer for a thorough review and for posing many important questions. We have tried to respond to each question/statement and have also edited the revised manuscript. Please find below a detailed point-by-point response to all comments (reviewers’ comments in black, our replies in blue).

Review Report 1

In this manuscript, the authors reported the synthesis of copper ferrite nanoparticles using anion exchange resin. Polyglucan and polyfructan with a molar mass from 5000 to 70000 Da were used to disperse nanoparticles, and their effect on the product yield and nanoparticle size was studied. In addition, the influence of nanoparticle size, annealing temperature, cooling rate, and elemental composition on the cubic phase stability was also studied. they also studied the effect of the CuFe2O4 nanoparticles structure on their magnetic properties. It is certain that the report would attract researchers in related research fields. Thus, I suggest the acceptance of the manuscript after solving these major problems.

  1. It is hard to recognize the lines in Figure 1, the authors should use different colors to distinguish the different lines.

Reply: Thank you, we have corrected that.

  1. In Figures 2 and 6, the PXRD pattern of CuFe2O4 should also be presented, as in Figure 8. Then it would be easier to compare the matching degree of the products with CuFe2O4.

Reply: Thank you very much for your comment. Rietveld refinement of the XRD pattern of the samples is added in the Supplementary Material (Figures S1,S3).

  1. It is clear that the absorption bands at 2853 and 2923 cm–1 of the sample obtained with dextran-70 are not negligible. The peaks can be clearly seen in Figure 3a, please revise it.

Reply: Thank you, we have corrected that.

  1. All the figures should be improved, including the definition and aesthetic.

Reply: Thank you, we have corrected that. We are thankful for your suggestion.

  1. In 3.2.1, the cubic phase contents are not only affected by the particle size but also the polysaccharide. It is better to use the same polysaccharide to synthesize particles of different sizes and then compare their cubic phase contents. 

Reply: We agree that it is better to use the same polysaccharide to synthesise different particle sizes and then compare the cubic phase content. However, particle size is determined by the nature of the polysaccharide. We cannot get particles of different sizes with the same polysaccharide. In addition, our experiments (see Figure 7, Table S1 for Supplementary Material, Figures S4, S5) indicate that the addition of the polysaccharide does not significantly affect the stability of the cubic copper ferrite.

The authors mentioned many times about supporting information, but it can not be found in the submitted files.

Reply: Thank you, we have corrected that.

Reviewer 2 Report

Manuscript ID: materials-2276604

Title: Copper ferrite nanoparticles synthesized using anion exchange resin: Influence of synthesis parameters on the cubic phase stability

Authors:  Saikova et al.

In this manuscript, authors have proposed a new simple and convenient method for the synthesis of copper ferrite (CuFe2O4) nanoparticles using a strongly basic anion exchange resin in OH form.

In my opinion, the work in this manuscript is original and well presented. However, there few things that must be done to recommend this paper for publishing as follow:

1-     The English language through whole manuscript should be modified and corrected.

2-     All figures should be carefully modified to have better quality.

3-     Some legends are missing in some figures.

4-     Results of this work should be compared with other reported results

Author Response

Reply: We are thankful to Reviewer for the careful reading of the manuscript and the constructive comments which allow us to improve the paper. Please find below a detailed point-by-point response to all comments

In this manuscript, authors have proposed a new simple and convenient method for the synthesis of copper ferrite (CuFe2O4) nanoparticles using a strongly basic anion exchange resin in OH form.

 In my opinion, the work in this manuscript is original and well presented. However, there few things that must be done to recommend this paper for publishing as follow:

  • The English language through whole manuscript should be modified and corrected.

Reply:  Thank you for the comment. The language in the manuscript has been modified and corrected. (Versions 2, 3)

  • All figures should be carefully modified to have better quality.

Reply: We are thankful for your suggestion. The figures have been revised. (Versions 2)

  • Some legends are missing in some figures.

Reply: The figures have been revised. (Versions 2)

  • Results of this work should be compared with other reported results

Reply: The results of this work are compared with other reported results. So, in section 3.2.1 we had shown that the stability of a cubic spinel phase at room temperature is not related to the size of the CoFe2O4 nanoparticles. The observed results are in agreement with those obtained in [21,48,49]. (The numbering given here corresponds to the numbering of the references in the manuscript). In [46] the sol-gel combustion method was used to obtain c-CuFe2O4 with particle size >100 nm. Submicron particles of stable cubic copper ferrite (Fd-3m) were synthesized under standard conditions by co-precipitation technique [21]. In addition, the tetragonal phase of copper ferrite can also contain small particles. For example, t-CuFe2O4 with a particle size of 15-25 nm has been reported [48, 49]. At the same time, some papers [43-47] suggest that the main factor in the stability of the cubic modification of copper ferrite is the particle size. The particles with size less than 40 nm contribute to the cubic phase stabilization, whereas larger submicron and micron particles lead to the formation of tetragonal copper ferrite.

In section 3.2.2 we established that there is no significant effect of polysaccharide additives and the annealing temperature in the range of 800-950 °C on the stability of c-CuFe2O4. While other studies [45, 55] confirm our findings, the references [29, 54] show that the high annealing temperature contributes to the formation of the tetragonal CuFe2O4 and the structural change c-CuFe2O4 →t-CuFe2O4 occurs as the annealing temperature increases.

In section 3.2.3 we have shown that the relative content of the cubic phase in the samples increases significantly when the samples are rapidly cooled in air (quenched).  These results are in agreement with literature data [46, 50, 51]. We also conclude that the stabilization of the cubic phase of copper ferrite is caused by the loss of oxygen during the high-temperature heat treatment of the precursors. The oxygen diffusion depends on the duration of the high temperature treatment of copper ferrite, which is not considered in many studies and may be the cause of the controversial results reported previously.

In addition, in the section 3.3 we compare the magnetic parameters of copper ferrite nanoparticles measured here (Ms = 34.6; 36,9 emu/g, Hc = 417; 420 Oe) with those reported elsewhere: Ms = 31.4 emu/g, Hc = 400.3 Oe [20], Ms = 27.4 emu/g, Hc = 526.6 Oe [21], Ms = 41.1 emu/g, Hc = 241 Oe [66], Ms = 32.4 emu/g, Hc = 517 Oe [67].

Reviewer 3 Report

This article describes a method for the production of copper ferrites based on the precipitation by an ion exchange resin of iron and copper cations in the presence of polysaccharides. This original method makes it possible, after heat treatment, to obtain nanometric spinel ferrites with low size dispersion. The article tends to prove that the presence of a cubic spinel phase at room temperature is not related to the size of the particles, nor to the addition of polysaccharides, but to the cooling rate and the cation content of the iron. These conclusions of the article are well supported by the experimental results.

Several remarks intended to improve the article can however be made.

Copper spinel ferrites and especially CuFe2O4, have been the subject of a large number of works spread over a fairly long period of time. Most of the bibliographical references, which are quoted, are however posterior to 1998, omitting in that many basic articles whose results were not fundamentally called into question. It would therefore be appropriate to cite these articles on which the more recent articles, that are referenced in the proposed paper, are based. Among these articles, we could mention in particular for the production methods and the structural aspects of copper ferrites:

- G.I. Finch et al. 1957 Proc. Royal Soc. London A Math. Phys. Science. 242, 28

- P. Wojtowicz 1959 Phys. Rev. 116, 32

- H.M. O'Bryan et al. 1966 J.Appl. Phys 37, 3, 1437.

- H. Ohnishi et al. 1961 J. Phys. Soc. Jpn 16, 1, 35 and 1959, J. Phys. Soc. Jpn 14, 106.

- L. Néel 1950 C.R. Acad. Science. Paris 190

- C. Villette et al. 1995 J. Solid State Chem. 117, 64 and 1998 J. Solid State Chem. 141, 56.

...

It has in particular been shown in these articles that a heat treatment above approximately 710° C causes the onset of CuO precipitation. The temperatures used in the work presented being much higher, it would be appropriate to comment more precisely on the effects of such heat treatments. It would also be good to enlarge figure 6 and indicate that in addition to sample 2, sample 4 also contains CuO.

It would also be clearer to specify that the products obtained after the precipitation are not spinel ferrites, but probably amorphous hydroxides. Otherwise, proof of the existence of a spinel phase must be provided by a relevant characterization technique (IR or Raman spectrometry, magnetic measurements, etc.?).

In calculating the magnetic moment of spinel ferrites using Néel's colinear model, the spin moment is usually taken into account. The ferric ions are thus assigned a moment of 5 Bohr magneton and the cupric ions a moment of 1. The values of 5.92 and 1.73 therefore do not seem relevant in this case.

It would be better to specify that the quadratic deformation of the lattice is due to a cooperative Jahn-Teller effect. The Jahn-Teller effect still exists in cubic copper ferrite, it is because it is no longer cooperative that the lattice is not deformed. The effect of the structural anisotropy, brought by the deformation of the lattice by cooperative Jahn-Teller effect, was demonstrated well before the article [51] quoted in reference. See in particular the work of Villette et al. cited above or F. Agnoli et al. 1999, C.R. Acad. Science. Paris 2, II, 525. It would be good not to forget to cite the pioneering articles in this field.

Finally, it seems that there is an error line 511. It is probably not cobalt ferrite, but copper ferrite.

From the point of view of the referee, the proposed article should therefore be subject to corrections and be completed in terms of bibliography, before publication.

Author Response

Reply: We are grateful to the reviewer for a thorough review and for posing many important questions. We have tried to respond to each question/statement and have also edited the revised manuscript. Please find below a detailed point-by-point response to all comments (reviewers’ comments in black, our replies in blue).

Review Report 2

This article describes a method for the production of copper ferrites based on the precipitation by an ion exchange resin of iron and copper cations in the presence of polysaccharides. This original method makes it possible, after heat treatment, to obtain nanometric spinel ferrites with low size dispersion. The article tends to prove that the presence of a cubic spinel phase at room temperature is not related to the size of the particles, nor to the addition of polysaccharides, but to the cooling rate and the cation content of the iron. These conclusions of the article are well supported by the experimental results.

Several remarks intended to improve the article can however be made.

Copper spinel ferrites and especially CuFe2O4, have been the subject of a large number of works spread over a fairly long period of time. Most of the bibliographical references, which are quoted, are however posterior to 1998, omitting in that many basic articles whose results were not fundamentally called into question. It would therefore be appropriate to cite these articles on which the more recent articles, that are referenced in the proposed paper, are based. Among these articles, we could mention in particular for the production methods and the structural aspects of copper ferrites:

- G.I. Finch et al. 1957 Proc. Royal Soc. London A Math. Phys. Science. 242, 28

- P. Wojtowicz 1959 Phys. Rev. 116, 32

- H.M. O'Bryan et al. 1966 J.Appl. Phys 37, 3, 1437.

- H. Ohnishi et al. 1961 J. Phys. Soc. Jpn 16, 1, 35 and 1959, J. Phys. Soc. Jpn 14, 106.

- L. Néel 1950 C.R. Acad. Science. Paris 190

- C. Villette et al. 1995 J. Solid State Chem. 117, 64 and 1998 J. Solid State Chem. 141, 56.

Reply: We are thankful for your suggestion. The bibliography has been expanded.

It has in particular been shown in these articles that a heat treatment above approximately 710° C causes the onset of CuO precipitation. The temperatures used in the work presented being much higher, it would be appropriate to comment more precisely on the effects of such heat treatments.

Reply: Thank you for the remark. The formation of CuO occurs during the thermal decomposition of CuFe2O4-δ in the temperature range 900-1100 °C: CuFe2O4-δ = 4δCu0.5Fe2.5O4 + (1-5δ)CuFe2O4+ 3δCuO

It would also be good to enlarge figure 6 and indicate that in addition to sample 2, sample 4 also contains CuO.

Reply: Thank you, we have corrected that. Rietveld refinement of the XRD pattern of the samples is presented in the supplementary material (Figure S3(d))

It would also be clearer to specify that the products obtained after the precipitation are not spinel ferrites, but probably amorphous hydroxides. Otherwise, proof of the existence of a spinel phase must be provided by a relevant characterization technique (IR or Raman spectrometry, magnetic measurements, etc.?).

Reply: Thank you very much for your comment. The corresponding remark was added to the manuscript.

In calculating the magnetic moment of spinel ferrites using Néel's colinear model, the spin moment is usually taken into account. The ferric ions are thus assigned a moment of 5 Bohr magneton and the cupric ions a moment of 1. The values of 5.92 and 1.73 therefore do not seem relevant in this case.

Reply: We are thankful for your suggestion; we have corrected that.

It would be better to specify that the quadratic deformation of the lattice is due to a cooperative Jahn-Teller effect. The Jahn-Teller effect still exists in cubic copper ferrite, it is because it is no longer cooperative that the lattice is not deformed. The effect of the structural anisotropy, brought by the deformation of the lattice by cooperative Jahn-Teller effect, was demonstrated well before the article [51] quoted in reference. See in particular the work of Villette et al. cited above or F. Agnoli et al. 1999, C.R. Acad. Science. Paris 2, II, 525. It would be good not to forget to cite the pioneering articles in this field.

Reply: Thank you, we have corrected that. We are thankful for your suggestion.

Finally, it seems that there is an error line 511. It is probably not cobalt ferrite, but copper ferrite.

Reply: Thank you, we have corrected that.

From the point of view of the referee, the proposed article should therefore be subject to corrections and be completed in terms of bibliography, before publication.

Round 2

Reviewer 1 Report

The revised paper basically solves the problems found in the first version, Thus I suggest the acceptance of the manuscript.

Author Response

Thank You for Your opinion

Reviewer 3 Report

From the referee's point of view, the article can now be published as is.

Author Response

Thank You for Your opinion